# Strategies to Name Metallo-β-Lactamases and Number Their Amino Acid Residues

**DOI:** 10.3390/antibiotics12121746

**Published:** 2023-12-16

**Authors:** Peter Oelschlaeger, Heba Kaadan, Rinku Dhungana

**Affiliations:** 1Department of Biotechnology and Pharmaceutical Sciences, College of Pharmacy, Western University of Health Sciences, Pomona, CA 91766, USA; heba.kaadan@westernu.edu (H.K.);; 2Department of Biological Sciences, Kenneth P. Dietrich School of Arts & Sciences, University of Pittsburgh, Pittsburgh, PA 15260, USA

**Keywords:** β-lactamase, metallo-β-lactamase, class B, naming, standard numbering

## Abstract

Metallo-β-lactamases (MBLs), also known as class B β-lactamases (BBLs), are Zn(II)-containing enzymes able to inactivate a broad range of β-lactams, the most commonly used antibiotics, including life-saving carbapenems. They have been known for about six decades, yet they have only gained much attention as a clinical problem for about three decades. The naming conventions of these enzymes have changed over time and followed various strategies, sometimes leading to confusion. We are summarizing the naming strategies of the currently known MBLs. These enzymes are quite diverse on the amino acid sequence level but structurally similar. Problems trying to describe conserved residues, such as Zn(II) ligands and other catalytically important residues, which have different numbers in different sequences, have led to the establishment of a standard numbering scheme for BBLs. While well intended, the standard numbering scheme is not trivial and has not been applied consistently. We revisit this standard numbering scheme and suggest some strategies for how its implementation could be made more accessible to researchers. Standard numbering facilitates the comparison of different enzymes as well as their interaction with novel antibiotics and BBL inhibitors.

## 1. Introduction

β-Lactam antibiotics have been the mainstay of antibacterial chemotherapy for eight decades. After the discovery of the antibacterial activity of penicillin by Alexander Fleming in 1928, it took more than a decade to bring it into the clinic [1,2,3,4]. Since then, penicillins and other subsequently discovered β-lactams, such as cephalosporins and cephamycins, carbapenems, and monobactams, have constituted more than half of all antibiotics used [5,6]. They owe their antibacterial effect to their ability to inhibit the bacterial enzyme transpeptidase, which is essential for the biosynthesis as well as the remodeling and repair of peptidoglycan, the major component of the bacterial cell wall. In addition, they are characterized by low toxicity because humans do not have a functional analog of this enzyme that could cause off-target effects, and most β-lactams are not metabolized and excreted renally. As predicted by Fleming in his Nobel lecture [7], resistance to β-lactams was discovered even before penicillin was used on a large scale in the 1940s [8], and inactivation of β-lactams by β-lactamases remains the most common antibiotic resistance mechanism [9].

Mechanistically, there are two types of β-lactamases. The first type uses an active-site serine residue to hydrolyze the antibiotic. These enzymes are referred to as serine β-lactamases or SBLs and constitute class A, C, and D β-lactamases, which are based on sequence homology. The second type activates a water molecule by coordination to Zn(II) ions, leading to deprotonation of the water molecule. The resulting hydroxide then acts as the nucleophile in the hydrolysis of the antibiotic. These enzymes are called metallo-β-lactamases or MBLs, constitute class B β-lactamases, and differ from the other three classes A, C, and D in structure, mechanism, substrate spectrum, and sensitivity to β-lactamase inhibitors, as described in the following.

The overall protein architecture of MBLs follows an αβ/βα fold [10] (Figure 1A).

A central “sandwich” of two β sheets is flanked by α helical domains on either side. In contrast, SBLs have a mixed αβ hydrolase fold. The catalytic mechanism of MBLs has been reviewed elsewhere (e.g., Bahr et al. [16] and references therein). It relies on one or two Zn(II) ions that activate the β-lactam substrate by polarizing the β-lactam carbonyl group, making the carbonyl carbon more electrophilic, and generating and orienting a hydroxide anion as an effective nucleophile. The nucleophilic attack of the hydroxide on the β-lactam carbonyl carbon initiates the hydrolysis of the β-lactam. SBLs employ an active-site serine residue to carry out the nucleophilic attack. In contrast to MBLs, the catalytic cycle of SBLs includes a covalent acyl-enzyme intermediate. Most MBLs, especially the clinically important ones, have a broad substrate spectrum that includes penicillins, cephalosporins and cephamycins, and carbapenems, but not monobactams. The substrate spectra of the other classes are diverse. Most of them can inactivate early penicillins and cephalosporins, and typically only enzymes that have acquired mutations (e.g., in the TEM family from class A) can hydrolyze more recent penicillins, cephalosporins, and monobactams [17]. In addition, there are specialized families (e.g., KPC, class A) that can inactivate carbapenems, and enzymes from other families (e.g., OXA, class D) have acquired mutations that enable them to hydrolyze carbapenems [18]. A clinically very significant difference between MBLs and SBLs is their sensitivity to β-lactamase inhibitors (BLIs). The original BLIs (clavulanate, sulbactam, and tazobactam) are themselves β-lactams, but they have no or very little antibacterial activity. These BLIs can inactivate some but not all SBLs and are ineffective against MBLs [19]. The discovery and development of BLIs, including MBL inhibitors, is an active field of research in both academia and the industrial sector, and recent progress has been summarized elsewhere [20,21,22,23,24].

Typical bacteria that express β-lactamases, including MBLs and SBLs, are Gram-negative Enterobacteriaceae (e.g., *Escherichia coli* and *Klebsiella pneumoniae*) and nonfermentative species, such as *Pseudomonas aeruginosa*, *Acinetobacter baumannii*, and *Stenotrophomonas maltophilia*. However, enzymes have been isolated from various other bacterial species, which are listed for each enzyme, for instance, in the β-Lactamase DataBase (BLDB) [25]. 

Accurate naming of MBLs and numbering of their amino acid residues is essential to understanding the function of MBLs and interpreting the binding and inhibition modes of potential MBL inhibitors. However, to quote Tooke et al. [20], “It is apparent […] that the field is considerably complicated by historical but now well-established inconsistencies of nomenclature”. This review offers an account of some of these historical naming and numbering conventions and some possible solutions to standardize enzyme names as well as amino acid residue numbering.

## 2. The Challenge of Naming β-Lactamases and Numbering Their Amino Acid Residues

In the first part of this review (Section 3), we will summarize different strategies that have been followed for naming metallo-β-lactamases and members within families. Family members are often numbered sequentially according to the chronology in which they were discovered, typically from clinical isolates. But, as can be expected, there have been controversies and miscommunications regarding who has the authority to make such assignments and which enzyme should have a particular name. Since these numberings have become part of the enzyme names (e.g., IMP-1, VIM-2, or NDM-1), we will consider these numbers part of the naming problem.

Another level of complexity is associated with the numbering of amino acid residues in β-lactamase sequences. Once it became clear that many enzymes were very closely related in terms of structure, including catalytic residues, the utility of standard numbering schemes became obvious. Such standard numbering schemes typically assign a constant number to an important catalytic residue. For instance, the active-site serine in class A and D enzymes is always S70 [26,27], and in class C enzymes, it is S64 [28]. The numbering of the other residues in the amino acid sequence is shifted accordingly. Thus, every amino acid in TEM-1 receives a number augmented by 2 [29] relative to its position in the preprotein (including the leader sequence) [26]. In the case of class A enzymes, this approach is straight-forward because sequences are overall very similar, typically resulting in the omission of a few residue numbers for some enzymes [29]. In MBLs (class B enzymes), the situation is more complex because the nucleophile is not one active-site serine, but an activated water/hydroxide coordinated to one or two Zn(II) ions, which in turn are coordinated by a set of amino acid ligands. The identity of these ligands mostly falls into three patterns, and based on these patterns, three subclasses of class B enzymes named B1, B2, and B3 have been defined [14] (Figure 1B–D). There have been efforts to number these Zn(II) ligands consistently [14,15,30]. Like the naming problem, there have been inconsistencies in the numbering of β-lactamase residues, obviously before standard numbering schemes were available but also after and despite their existence. 

The simpler it is to conform to a particular numbering scheme, the more consistently it is followed. For instance, as mentioned above, most class A amino acid sequences can be brought into agreement with Ambler’s numbering scheme quite easily. The metallo- or class B β-lactamase (BBL) standard numbering scheme [14,15] is quite complex, with various possible insertions and deletions. Accordingly, when new enzymes were discovered, BBL numbering was not always applied. Whether the BBL standard numbering is used for PDB files derived from X-ray crystal structures or not often seems to depend on whether existing PDB files that are used as search models have used BBL numbering or not. The same is true for publications: if most publications about an enzyme have not used a standard numbering scheme, then most subsequent publications will not use it either.

In the second part of this review (Section 4), we will revisit the BBL standard numbering scheme, explain the potential benefits of using it, give a historic overview of when it has been applied and when not, and provide possible solutions for how its use could be increased without imposing an undue burden on researchers.

## 3. Naming of β-Lactamases

### 3.1. Naming of β-Lactamases Based on the Enzymatic Reaction Catalyzed

The enzyme name β-lactamase refers to the activity of hydrolyzing the β-lactam ring of these antibiotics (Figure 2) and is classified by the Enzyme Commission as EC 3.5.2.6 (spelled out: 3 = hydrolases, 5 = acting on carbon-nitrogen bonds other than peptide bonds, 2 = in cyclic amides, and 6 = β-lactamase) [31,32].

It should be noted that other hydrolases exist that act on β-lactam antibiotics outside the β-lactam ring. These include natural or engineered penicillin and cephalosporin amidases (EC 3.5.1.11, 3 = hydrolases, 5 = acting on carbon-nitrogen bonds, 1 = in linear amides, and 11 = penicillin amidase) [33]. These enzymes are historically also referred to as acylases. They are frequently used in biotransformations to cleave the amide bond at C6 of penicillins or C7 of cephalosporins to yield 6-aminopenicillanic acid or 7-cephalosporanic acid, respectively. The same enzymes can be used subsequently to introduce more desirable moieties at those positions to yield novel semisynthetic β-lactam antibiotics. These processes are described in detail elsewhere [34,35,36]. Beyond that, these amidases/acylases are also used in other biocatalytic processes, such as peptide synthesis and the production of enantiopure compounds [37].

Another hydrolytic reaction that is involved in the degradation of cephalosporins with an ester moiety at position 3 (3-acetoxymethylcephalosporins), such as cephalothin and cefotaxime, is ester hydrolysis in either acidic or basic conditions to the desacetyl derivatives. The resulting alcohols can form lactones with the carboxyl group at C4 in acidic conditions [38]. The desacetyl derivatives do maintain antimicrobial activity [39]. This reaction is acid- or base-catalyzed and does not require an enzyme.

This review will focus exclusively on metallo-β-lactamases and hydrolysis of the β-lactam ring (EC 3.5.2.6). An interesting observation is that when the first β-lactamase activity was reported, it was not at all clear that it was a β-lactamase. This is evident from the title of the seminal publication about an “enzyme […] able to destroy penicillin”, or the enzyme name penicillinase [8]. In fact, the structure of penicillin was not solved until the late 1940s by X-ray crystallography [40] and infrared spectroscopy [41]. Once the bioactive core of penicillin (as well as cephalosporins discovered later) was identified as a β-lactam ring and the inactivating enzymes were shown to hydrolyze the β-lactam ring, they could have properly been called β-lactam hydrolases, but since the mid-1960s to this day, usually the shorter and less specific term β-lactamase is used. Historical perspectives on β-lactamase nomenclature, such as naming enzymes based on their substrate preference, are summarized elsewhere [42].

Also in the mid-1960s, it became apparent that there are two mechanistically distinct types of β-lactamases: one that is not inactivated by the addition of the metal ion-chelating agent ethylenediaminetetraacetate (EDTA) and one that is [43,44]. The former was determined to be a class A serine β-lactamase (SBL), and the latter a class B (BBL) or metallo-β-lactamase (MBL). The SBL (a penicillinase in terms of substrate spectrum) and MBL (a cephalosporinase) from *Bacillus cereus* studied by Sabath and Abraham were subsequently simply referred to as *B. cereus* β-lactamase type I or BcI [45], and *B. cereus* β-lactamase type II or BcII [46]. The reader might notice that naming two enzymes that are structurally and mechanistically so different by these names might cause confusion later. Very similarly, when two β-lactamases referred to as “labile enzymes” were isolated from *Stenotrophomonas maltophilia*, one that turned out to be an MBL was named L1 [47] and another one that turned out to be an SBL was named L2 [48]. 

Classification of β-lactamases is accomplished by amino acid sequence comparison [49] or by also including function and physical properties [9]. Historical, frequently confusing, [42,50] and desirable [51] naming of all β-lactamases has been described. Here, the focus is on class B or MBLs. For an in-depth classification of all β-lactamases, including SBLs, we refer to other recent excellent reviews [9,16,20,42,52]. In terms of naming, the identity of MBLs is often indicated in the name by using the letter M to indicate MBL (for instance, in NDM for New Delhi MBL or VIM for Verona Integron-borne MBL) or B to indicate class B (for instance, in BlaB for β-lactamase class B or CGB for *Chryseobacterium gleum* class B), or the combination MB to indicate an MBL (for instance, in GMB for German MBL or HMB for Hamburg MBL). 

### 3.2. Naming of Metallo-β-Lactamase Families

Apart from some exceptions, many of which date back to the 20th century, most MBL families (as well as SBL families) are named by a three-letter acronym, but what the acronyms stand for is not consistent (Figure 3). Several databases containing β-lactamase information exist [25,53,54,55,56,57,58,59]. For this study, we have relied mostly on the β-Lactamase DataBase (BLDB) [25]. Table 1, Table 2 and Table 3 list the named MBLs found in the BLDB with their abbreviated name (as shown in the database), the derivation, naming pattern, naming pattern explanation, and reference (including PMID). We cross-checked our derivations with those reported by Dr. George Jacoby [50] where possible and contacted authors when in doubt. In some cases, we used our own best judgment.

Analyzing Table 1, Table 2 and Table 3, one can observe that the three-letter naming patterns used are very diverse. The most common are *Gs*M (*Genus species*
MBL), followed by *Ges* (*Genus species*), *Gen* (*Genus*), *Gsp* (*Genus species*), LSu (Location Substrate Specificity), and LoM (Location MBL). The different strategies for obtaining these three-letter codes are summarized in Figure 3. 

Either using *Genus* and/or *species* or a specific location in the name has its pros and cons. If a *Genus* and/or *species* are used, chances are that the enzyme is subsequently isolated from another organism or that the name of the organism changes due to reclassification. GOB-1 was isolated from *Chryseobacterium meningosepticum,* with the GO derived from the center of the species name and B added for class B or BBL. Subsequent variants were isolated from other organisms, like *Elizabethkingia anopheles,* making the acronym somewhat meaningless. In addition, an increasing number of β-lactamase genes are isolated from metagenomes where *Genus* and *species* are frequently not known. Regarding enzymes that use a location in their name, the first Verona integron-borne MBL (VIM-1) was isolated in Verona, Italy [97], but VIM-2 was isolated in Marseille, France [128]. In addition, giving an MBL a name associated with a specific location can be politically sensitive [129]. A recent contribution by a panel of β-lactamase experts recommended that “new β-lactamases should not be named based on geographical location” [51]. Naming an enzyme based on substrate preference is equally problematic, since frequently they turn out to have an even greater preference for other substrates. For instance, IMP-1 was initially named for its ability to inactivate imipenem [81], although it is actually much more efficient at inactivating some cephalosporins [130,131]. Even when only focusing on carbapenemase activity, a variant named IMP-6 is particularly efficient at inactivating meropenem [132,133]. Nevertheless, its name is still IMP-6 rather than MER or MEM for meropenemase.

The authors do not have any specific recommendations on how to name new MBLs except that, if possible, a combination of three capital letters should be used that is somewhat descriptive of the enzyme’s origin or properties and ideally ends in M for MBL. There is a large body of scientific studies published on existing MBLs based on the established nomenclature. So, we should embrace this historically grown nomenclature and attempt to avoid ambiguity going forward. Authors should also consult Ref. [51] and are encouraged to contact NCBI staff as suggested here: https://www.ncbi.nlm.nih.gov/pathogens/submit-beta-lactamase/ (accessed on 31 October 2023). 

### 3.3. Naming of Metallo-β-Lactamase Family Members

Historically, when a new MBL was discovered that was deemed sufficiently different from other known MBL families, a new name was given as described above (preferably a capital three-letter name) with the number 1 added (e.g., IMP-1 [81], VIM-1 [97], and NDM-1 [86]). As additional variants of these enzymes or their encoding genes were discovered, new allele numbers 2, 3, etc. were added to the family acronym in chronological order. These enzymes, with links to the original publication, the nucleotide sequence, as well as the amino acid sequence, were then deposited on the “Lahey Site”, which was curated for many years by Drs. Karen Bush, George Jacoby, and later Timothy Palzkill. As was recently explained [42], this site was retired in 2015 but is still available in its 2015 version at https://externalwebapps.lahey.org/studies/Other.aspx (accessed on 31 October 2023). The authors of that site also had the authority to assign/approve new family names and new allele numbers to existing families. Today, the information on the Lahey Site as well as the name-giving authority reside with the National Center for Biotechnology Information (NCBI), Bethesda, MD, USA, for instance, at the National Database of Antibiotic Resistant Organisms (NDARO) (https://www.ncbi.nlm.nih.gov/pathogens/antimicrobial-resistance/, accessed on 31 October 2023). 

Such an allele assignment system is not perfect. As mentioned, allele numbers were assigned in chronological order, but typically this does not mean isolation date but publication date (or rather, allele number request date). For example, VIM-2 was isolated in 1996 [128] before VIM-1 in 1997 [97], but published in 2000 after the publication of VIM-1 in 1999. In addition, while the proximity of allele numbers might be interpreted as an indication of sequence similarity, this is far from the truth. IMP-1 isolated in Japan [81] and IMP-2 isolated in Italy [134] are among the most distantly related IMP enzymes with only 85% sequence identity. Interestingly, IMP-2 was isolated from a patient in Verona, Italy, and it was integron-borne, but it was not named Verona integron-borne MBL (VIM)-2 but instead IMP-2 due to its higher sequence similarity with IMP-1 than VIM-1 (31%), further highlighting some of the curiosities of MBL naming. Nevertheless, this naming approach has resulted in close to 200 BcII variants, more than 100 IMP variants, and approaching 100 VIM and NDM variants. For some of the SBL families, the allele numbers are even larger (>2200 EC variants (class C) and >1200 OXA variants (class D)) [25].

Unfortunately, authors reporting new families or new alleles did not always consult with the authorities mentioned above, leading to incorrect assignments and duplications of assignments. For instance, the name NDM-16 was used twice for different enzymes, which differ in four amino acid identities. This necessitated the designation of the two names, NDM-16a (GenBank accession code NG_049333) and NDM-16b (GenBank accession code KU285430). Another example is the designation of two different names, IND-2a (GenBank accession code AF219130) and IND-13 (GenBank accession code HM245381), for the identical enzyme.

Again, the authors recommend consulting Ref. [51] and contacting NCBI staff to request the assignment of a new allele number before publishing any such enzyme. More information can be found here: https://www.ncbi.nlm.nih.gov/pathogens/submit-beta-lactamase/ (accessed on 31 October 2023).

## 4. Numbering Amino Acid Residues in Metallo-β-Lactamases

Now that each enzyme is assigned a unique identifier through its family name and number, what remains to be labeled in a way that is useful to microbiologists, biochemists, and medicinal chemists are the amino acid residues. Because β-lactamases from each class are quite diverse in amino acid sequence with insertions and deletions but very similar in three-dimensional structure with conserved residues, it has been suggested to apply standard amino acid numbering schemes based on sequence and structural alignments. Such standard numberings now exist for all classes: A [29], B [14,15], C [28], and D [27]. The one for class B enzymes will be revisited below. 

### 4.1. The Class B (Metallo-)β-Lactamase Standard Numbering Scheme

In 2001, several experts in the MBL field with backgrounds in enzymology and X-ray crystallography proposed a “standard numbering scheme for class B β-lactamases”, henceforth often referred to as the BBL or MBL standard numbering scheme [14]. At that time, the crystal structures of a few MBLs were available. These as well as sequence similarities were the basis of grouping MBLs or class B enzymes into three subclasses, B1–B3. B1 enzymes included BcII, CcrA, and IMP-1, for which crystal structures were available, and VIM-1, BlaB, and IND-1, for which no structures were available at the time. One crystal structure of a B3 enzyme (L1, Figure 1D) was available, and other B3 enzymes included in the study were FEZ-1, GOB-1, and THIN-B. The authors noted that, despite the small sequence identities of the different enzymes, their overall structures were very similar. This information was sufficient to align the amino acid sequences from all three subclasses, including B2 enzymes CphA and Sfh-1, for which no structures were available while keeping the positions of Zn(II) ligands constant. The N-terminal numbering of L1 was used as a reference due to it being the longest enzyme. However, there are some deletions from residues 58–65 due to B3 enzymes not having an active site lid. Aligning the Zn(II) ligands based on their three-dimensional location also made it clear that they deviate between the subclasses, one defining characteristic of the subclassification (Table 4 and Figure 1). All MBLs have two Zn(II) binding sites. The Zn(II) ions bound are called Zn1 and Zn2, and their respective ligands are referred to as Zn1 ligands and Zn2 ligands.

A few years later, additional crystal structures were available for VIM-2 and BlaB (subclass B1), FEZ-1 (subclass B3), and importantly, CphA as the first subclass B2 enzyme (Figure 1C), which prompted an update to the BBL standard numbering scheme [15]. The identity of the Zn(II) ligands did not have to be adjusted, and minor adjustments to the numbering were mostly limited to the N- and C-termini. The CphA crystal structure also revealed that while B1 and B3 enzymes bind two Zn(II) ions, B2 only binds Zn2 [11], and Zn1 inhibits B2 enzymes. A later crystal structure confirmed that the inhibitory Zn1 binds to the Zn1 binding site [135]. The decreased affinity for Zn1 could be explained by N116 instead of H116 in the Zn1 site (Figure 1C). Table 4 also shows that subclass B3 is the most diverse in terms of Zn(II) ligands, with some of these variants being described only recently in a genome database [112] and the E116 variant subsequently studied biochemically [124]. Following the publication of the BBL standard numbering scheme, many researchers in the field made an effort to apply it in publications and even in PDB files for all hitherto-crystallized B2 [11,12,135,136,137,138,139,140] and many B3 enzymes [12,141,142,143,144,145,146,147]. 

Enter NDM-1 [86]. NDM-1 caused severe outbreaks and quickly became one of the most widely spread MBLs and antibiotic resistance factors [148]. It was also featured in a PBS Frontline documentary called “Hunting the Nightmare Bacteria”. Understandably, there was a big desire to learn more about this enzyme, including solving a crystal structure. Eventually, it was solved by research groups that either decided not to use the BBL numbering or were unfamiliar with it [149,150,151]. Going forward, most publications on NDM-1 have not used the BBL numbering, with a few laudable exceptions [125,152,153,154].

Of course, the original motivation for the BBL numbering scheme is still valid. If applied consistently, it would greatly improve our ability to compare different MBL variants, their catalytic mechanisms, and their interactions with substrates as well as inhibitors. We propose that the reasons for not applying it are mostly unfamiliarity with its existence and how to apply it. Indeed, its application is not trivial, and, to our knowledge, no easily accessible tool exists. We have previously proposed a simple algorithm for renumbering residues in NDM-1 [30], but it requires manually renumbering residues by certain numbers depending on their position in the protein (Table 5). In the next section, we illustrate the renumbering problem with manual renumbering and then propose possible solutions for automated renumbering.

### 4.2. Strategies for MBL Renumbering

#### 4.2.1. Manual Renumbering

MBL amino acid sequences in public databases, such as NCBI Protein, are shown in FASTA format, that is, as strings of letters without any numbering. Three-dimensional structures, mostly from X-ray crystallography, are deposited as Protein Data Bank (PDB) files, and the amino acid numbering is often simply the position of the amino acid in the FASTA file. For instance, the NDM-1 preprotein sequence deposited under ID WP_004201164 is shown in Figure 4.

The most recent structure of NDM-1 has been deposited under PDB ID 8PGE (Figure 1A,B). It is in complex with hydrolyzed benzylpenicillin at 1.4 Å resolution and contains two chains. The BBL standard numbering has not been applied. The mature protein starts with C26, highlighted in blue, which is lipidated and anchors the protein into the bacterial outer membrane [151,155]. The PDB structure starts with I31 in both chains, which is expected because C26 is often removed to improve expression of soluble NDM-1 and/or a few residues at the N-terminus could be disordered in the crystal structure. Both chains end with the final residue, R270. The range covered by the PDB file is highlighted in green in Figure 4. The active-site Zn(II) ligands are highlighted in red. Table 5 shows the renumbering algorithm proposed previously [30], which, when applied to the sequence shown in Figure 4, yields the correct BBL numbers. For instance,

H120, H122, and D124 become H116, H118, and D120 (−4), respectively;H189 becomes H196 (+7);C208 becomes C221 (+13);H250 becomes H263 (+13).

This simple method has accurately renumbered all Zn(II) ligands and can renumber all other residues accordingly. However, this process may be too tedious for most researchers. In addition, it is limited to NDM family enzymes that have the same length as NDM-1. For instance, it will not work for NDM-18, which has a 5-amino acid residue duplication after its original position 46, BBL numbering 40. It would require changing the numbering of the following five residues to 40a through 40e.

#### 4.2.2. Automated Renumbering Based on Conserved Motive Recognition

A better approach than just using absolute numbers would be to search for conserved motives, such as the HXHXD motive (where X could be any amino acid), that is highly conserved in B1 and many B3 enzymes, and adjust the numbering in that region to the desired numbers: H116, X117, H118, X119, and D120. Subsequently, other motives in other parts of the protein can be used to adjust the numbering in those regions. A Python program doing just that was presented at the ASM Microbe 2022 [156]. This program can also account for the five amino acid duplication in NDM-18 mentioned above. Once determined, the BBL numbering can also be used to renumber amino acids in PDB files. Still, such a program is specific to one family and would have to be adapted to any other MBL family, which may not be very practical on a larger scale.

#### 4.2.3. Automated Renumbering Based on Profile Hidden Markov Model

Profile hidden Markov models have been used to renumber, for example, thiamine diphosphate-dependent decarboxylases [157] or class C β-lactamases [28]. This approach is more biologically sound, as it does not require the presence of specific consensus sequences and can thus cover a broader range of MBL sequences, including unknown ones. It involves a structure-guided alignment of multiple relevant sequences and the creation of a profile hidden Markov model, for instance, with the HMMER program (http://hmmer.org, accessed on 31 October 2023). Both the alignment and assignment of standard numbers can be adjusted to ensure agreement with the BBL standard numbering scheme [15].

Finally, for a tool to find wide use, it needs to be freely accessible and easy to use. Mack et al. provided detailed instructions on how to install and run HMMER with their profile hidden Markov model and how to interpret the output [28]. It may be even easier for a user to copy-and-paste or upload a FASTA or PDB file with original numbering and be returned a sequence file with renumbering (probably in a comma-separated value or CSV file format, showing original numbering, amino acid identity, and standard numbering) or a renumbered PDB file. This could all be accomplished through a web interface. Such a web interface has been implemented for thiamine diphosphate-dependent decarboxylases (https://teed.biocatnet.de/numbering/, accessed on 31 October 2023 [157]). We are in the process of developing a similar web site for renumbering MBLs.

## 5. Conclusions

The field of MBLs has grown exponentially over the past three decades, assisted by advances in DNA sequencing and X-ray crystallography technology and, of course, through the studies performed by dedicated clinicians and researchers. Unfortunately, the importance of MBLs as the cause of disease has also grown, probably illustrated mostly through the sudden appearance followed by severe outbreaks of NDM-1. Efforts to design new β-lactam antibiotics that cannot be inactivated by MBLs and MBL inhibitors are ongoing [16,20,22,158,159], and having sequence and structural information on an increasing number of MBL families and family members has been beneficial for these efforts. However, the more information we have, the more important it is to keep it organized. Just as one needs to keep their files organized on a computer by putting them into the appropriate folders and giving them descriptive and unique names, the classification and naming of MBLs have also become more challenging and important. Here, we have attempted to give an account of naming strategies to highlight this challenge. Some of them have historical origins, and we must embrace the names we have. Going forward, an expert panel has made some recommendations regarding β-lactamase nomenclature [51], and we wholeheartedly agree with those recommendations. Many of the recommendations relate to the assignment of new allele (family member) numbers. Besides, new families should not be named based on geographical location. We additionally recommend that a three-letter code with M in the third position be used for MBLs. This allows for 26 × 26 = 676 different combinations, a number that might be exceeded at some point. Then, probably the easiest way forward would be to move to a four-letter code.

Regarding the numbering of amino acid residues in MBLs, the best and most efficient path forward will likely involve the use of growing databases of MBL sequences and structures, automated sequence and structure alignments, and the automated assignment of amino acid numbers based on, for instance, profile hidden Markov models and the already established standard numbering scheme. These efforts will be facilitated by ever-increasing computational resources and improved computer algorithms. It is expected that this increased knowledge base will benefit the design and testing of new β-lactam antibiotics and MBL inhibitors and improve the treatment of antibiotic-resistant bacterial infections.

## Figures and Tables

**Figure 1 antibiotics-12-01746-f001:**
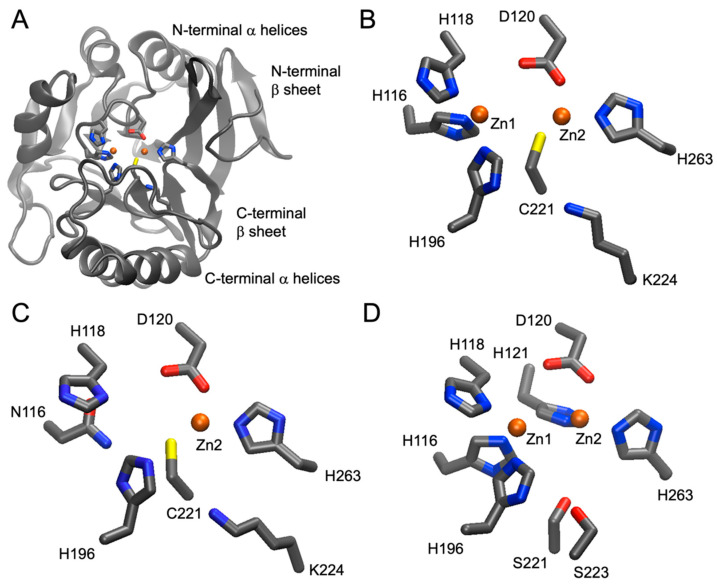
Three-dimensional structures of MBLs. (**A**) Overall structure of NDM-1 (subclass B1) based on PDB entry 8PGE. The protein backbone is shown in gray as a cartoon representation. The side chains of the Zn(II) ligands in the active site as well as K224, which is involved in substrate binding, are shown as sticks (C, gray; N, blue; O, red; and S, yellow). Zn(II) ions are shown as orange spheres. (**B**) A detailed view of the active site of NDM-1 is shown in panel (**A**), with residues labeled at their Cα atoms. (**C**) Active site of CphA (subclass B2) based on PDB entry 1X8G [11]. H116 in subclass B1 is replaced with N116, and Zn1 is absent. (**D**) Active site of L1 (subclass B3) based on PDB entry 2AIO [12]. The role of Zn(II) ligand C221 in subclasses B1 and B2 is taken over by H121. S221 and S223 are involved in substrate binding rather than K224. Figures were created with VMD [13]. The residues are numbered following the class B standard numbering scheme [14,15].

**Figure 2 antibiotics-12-01746-f002:**
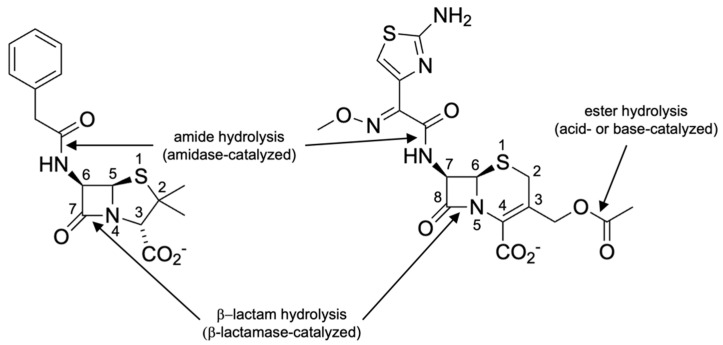
Chemical structures of benzylpenicillin (**left**) and cefotaxime (**right**) as representative β-lactam antibiotics. Different possible hydrolysis reactions acting on these compounds are shown with the catalyst (enzyme or acid/base) indicated.

**Figure 3 antibiotics-12-01746-f003:**
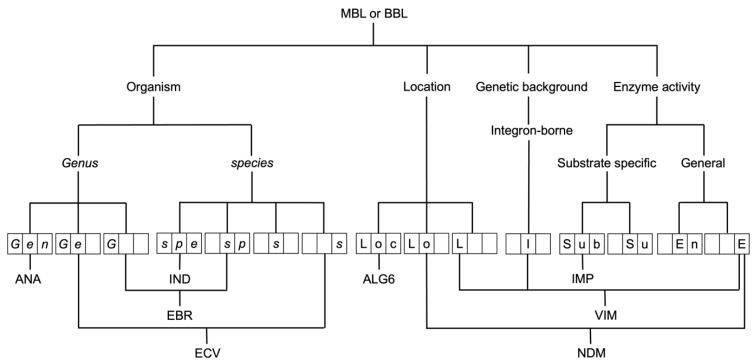
This decision tree chart shows the different strategies that have been applied in the naming of MBLs. In this chart, only three-letter-code names have been considered. The main themes by which names have been given are based on organism of origin, location where the enzyme was isolated, genetic background, such as integron-borne, and enzyme activity, either generally as MBL or class B or focusing on its substrate specificity, such as imipenemase. These different strategies have led to various three-letter-code names, the components of which are shown in the three-digit boxes. Below these boxes, some examples are shown of where these codes are used, either individually or in combination.

**Figure 4 antibiotics-12-01746-f004:**
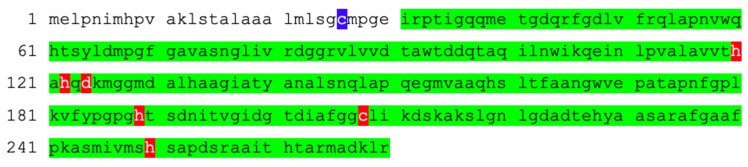
Amino acid sequence of the NDM-1 preprotein deposited in NCBI Protein under ID WP_004201164. The portion included in PDB entry 8PGE is highlighted in green. The first amino acid of the mature protein is highlighted in blue. Zn(II) ligands are highlighted in red.

**Table 1 antibiotics-12-01746-t001:** Naming of subclass B1 enzymes. The underlined letters in the Derivation column indicate where the letters in the Name column are derived from. The underlined letters in the Pattern Explanation column indicate where the letters in the Pattern column are derived from. The references are designated by their PubMed ID and their number in the References section.

Name	Derivation	Pattern	Pattern Explanation	Reference
AFM	*Alcaligenes faecalis* MBL	*Gs*M	*Genus species* MBL	36000902 [60]
ANA	*Anaeromyxobacter* spp.	*Gen*	*Genus*	29020980 [61]
BcII	*Bacillus cereus* type II BL	*Gs*	*Genus species*	3930467 [46]
BIM	Belém imipenemase	LSu	Location Substrate Specificity	37038995 [62]
BlaB	β-lactamase class B	*Other **		10858348 [63]
CAM	Central Alberta MBL	LoM	Location MBL	30789204 [64]
CfiA (CcrA)	Cefoxitin and imipenem-resistant A Cefoxitin and carbapenem-resistant A	*Other* *Other*		2110145 [65] 2121094 [66]
CGB	*Chryseobacterium gleum* class B	*Gs*B	*Genus species* BBL	12183230 [67]
CHM	*Chryseobacterium* MBL	*Ge*M	*Genus* MBL	37047008 [68]
CEMC19	Cefixime (cem) resistance	*Other*		35768448 [69]
CrxA	Carbapenem-resistant *Bacteroides xylanisolvens* A	*Other*		35296904 [70]
CX1	Isolated from clone CX1	*Other*		31487611 [71]
DIM	Dutch imipenemase	LSu	Location Substrate Specificity	20308383 [72]
EBR	*Empedobacter brevis*	*Gsp*	*Genus species*	12234848 [73]
ECV	*Echinicola vietnamensis*	*Ges*	*Genus species*	29020980 [61]
ElBla2	*Erythrobacter litoralis* β-lactamase 2	*Other*		21468894 [74]
FIA	*Fibrella aestuarina*	*Ges*	*Genus species*	29020980 [61]
FIM	Florence imipenemase	LSu	Location Substrate Specificity	23114762 [75]
GIM	German imipenemase	LSu	Location Substrate Specificity	15561840 [76]
GMB	German MBL	LMB	Location MBL	35257174 [77]
GRD23	*Gemmatimonadetes* resistant Denmark	*Other*		28082950 [78]
HBA	*Hirschia baltica*	*Gsp*	*Genus species*	22675580 [79]
HMB	Hamburg MBL	LMB	Location MBL	28065891 [80]
IMP	Imipenemase	Sub	Substrate Specificity	8141584 [81]
IND	*Chryseobacterium indologenes*	*spe*	*species*	10077836 [82]
JOHN	*Chryseobacterium johnsoniae*	*spec*	*species*	12562690 [83]
KHM	Kyorin University Hospital MBL	LoM	Location MBL	18765691 [84]
MOC	*Myroides odoratus* carbapenemase	*Gs*M	*Genus species* MBL	
MUS	*Myroides odoratimimus*	*spe*	*species*	12384365 [85]
MYO	*Myroides odoratimimus*	*Ges*	*Genus species*	29020980 [61]
MYX	*Myxococcus xanthus*	*Ges*	*Genus species*	29020980 [61]
NDM	New Delhi MBL	LoM	Location MBL	19770275 [86]
ORR	*Ornithobacterium rhinotracheale*	*Ges*	*Genus species*	29020980 [61]
PAN	*Pseudobacteriovorax antillogorgiicola*	*Gsp*	*Genus species*	31396187 [87]
PEDO	*Pedobacter roseus*	*Genu*	*Genus*	26482314 [88]
PKB	*Pontibacter korlensis* class B	*Gs*B	*Genus species* BBL	26057562 [89]
PST	*Pseudomonas stutzeri*	*Gsp*	*Genus species*	29020980 [61]
SFB	*Shewanella frigidimarina* class B	*Gs*B	*Genus species* BBL	15772146 [90]
SHD	*Shewanella denitrificans*	*Ges*	*Genus species*	29020980 [61]
SHN	*Shewanella denitrificans*	*Ges*	*Genus species*	29020980 [61]
SIM	Seoul imipenemase	LSS	Location Substrate Specificity	16251286 [91]
SLB	*Shewanella livinstonensis* class B	*Gs*B	*Genus species* BBL	15772146 [90]
SPM	Sao Paulo MBL	LoM	Location MBL	12407123 [92]
SPN79	Spain	Loc	Location	28082950 [78]
SPS	*Sediminispirochaeta smaragdinae*	*Ges*	*Genus species*	29020980 [61]
STA	*Stigmatella aurantiaca*	*Ges*	*Genus species*	29020980 [61]
SZM	*Shenzhen* MBL	LoM	Location MBL	36225370 [93]
TMB	Tripoli MBL	LMB	Location MBL	22290947 [94]
TTU	*Teredinibacter turnerae*	*Gsp*	*Genus species*	29020980 [61]
TUS	*Myroides odoratus*	*spe*	*species*	12384365 [85]
VAM (VMB)	*Vibrio alginolyticus* MBL *Vibrio alginolyticus* MBL	*Gs*M *G*MB	*Genus species* MBL *Genus* MBL	34424042 [95] 34097496 [96]
VIM	Verona integron-borne MBL	LIM	Location integron-borne MBL	10390207 [97]
VMB	*Vibrio* MBL	*G*MB	*Genus* MBL	32293144 [98]
VMH	*Vibrio vulnificus* metallohydrolase	*G*MH	*Genus* metallohydrolase	34228542 [99]
WUS	Wenzhou *Monopterus albus*	L*sp*	Location *species*	36532482 [100]
ZHO	*Zhongshania aliphaticivorans*	*Gen*	*Genus*	30778547 [101]
ZOG	*Zobellia galactanivorans*	*Ges*	*Genus species*	29020980 [61]

* *Other*, does not fit into one of the commonly used patterns.

**Table 2 antibiotics-12-01746-t002:** Naming of subclass B2 enzymes.

Name	Derivation	Pattern	Pattern Explanation	Reference
CphA	Carbapenem-hydrolyzing enzyme A	*Other*		1856163 [102]
CVI	*Chromobacterium violaceum*	*Gsp*	*Genus species*	37513808 [103]
PFM	*Pseudomonas fluorescens* MBL	*Gs*M	*Genus species* MBL	31685461 [104]
Sfh	*Serratia fonticola* carbapenem hydrolase	*Gs*H	*Genus species* hydrolase	12821491 [105]
YEM	*Yersinia mollaretii*	*Ges*	*Genus species*	32540974 [106]

**Table 3 antibiotics-12-01746-t003:** Naming of subclass B3 enzymes.

Name	Derivation	Pattern	Pattern Explanation	Reference
AIM	Adelaide imipenemase	LSu	Location Substrate Specificity	22985886 [107]
ALG6	Algeria	Loc	Location	28082950 [78]
ALG11	Algeria	Loc	Location	28082950 [78]
AM1	Isolated from clone AM1	*Other*		31487611 [71]
B3SU1	B3 subclass uncultured 1	*Other*		
B3SU2	B3 subclass uncultured 2	*Other*		
BJP	*Bradyrhizobium japonicum*	*Gsp*	*Genus species*	16723554 [108]
BLEG	*Bacillus lehensis* G	*Gsp*G	*Genus species*	34502284 [109]
CAR	*Erwinia caratovora*	*spe*	*species*	18443127 [110]
CAU	*Caulobacter crescentus*	*Gen*	*Genus*	12019096 [111]
CHI	*Chitinophaga pinensis*	*Gen*	*Genus*	
CPS	*Chryseobacterium piscium* Stok-1	*Gs*s	*Genus species* strain	26482314 [88]
CRD3	CRUCIAL Denmark	*Other*		28082950 [78]
CSR	*Chronobacter sakazakii* resistant	*Gs*r	*Genus species* resistant	32542533 [112]
DHT2	Dossenheim plantomycin treated	*Other*		28082950 [78]
EAM	*Erythrobacter aquimaris* MBL	*Gs*M	*Genus species* MBL	22850693 [113]
ECM	*Erythrobacter citreus* MBL	*Gs*M	*Genus species* MBL	22850693 [113]
EFM	*Erythrobacter flavus* MBL	*Gs*M	*Genus species* MBL	22850693 [113]
ELM	*Erythrobacter longus* MBL	*Gs*M	*Genus species* MBL	22850693 [113]
ESP	Extended-spectrum BL	*Other*		26482314 [88]
EVM	*Erythrobacter vulgaris* MBL	*Gs*M	*Genus species* MBL	22850693 [113]
FEZ	*Fluoribacter gormanii* endogenous zinc BL	*Other*		10817705 [114]
GOB	*Chryseobacterium meningosepticum* class B	*sp*B	*species* BBL	10858348 [63]
L1	Labile enzyme 1 from *Stenotrophomonas maltophilia*	*Other*		8018721 [47]
LMB	Linz MBL	LMB	Location MBL	29897538 [115]
LRA2	Lactam resistant from Alaskan soil	*Other*		18843302 [116]
LRA3	Lactam resistant from Alaskan soil	*Other*		18843302 [116]
LRA7	Lactam resistant from Alaskan soil	*Other*		18843302 [116]
LRA8	Lactam resistant from Alaskan soil	*Other*		18843302 [116]
LRA12	Lactam resistant from Alaskan soil	*Other*		18843302 [116]
LRA17	Lactam resistant from Alaskan soil	*Other*		18843302 [116]
LRA17	Lactam resistant from Alaskan soil	*Other*		18843302 [116]
MIM	Maynooth imipenemase	LSS	Location Substrate Specificity	26775612 [117]
MSI	*Massilia oculi*	*Gen*	*Genus*	26482314 [88]
NWM	North Rhine-Westphalia MBL	LoM	Location MBL	
PAM	*Pseudomonas alcaligenes* MBL	*Gs*M	*Genus species* MBL	24356301 [118]
PEDO	*Pedobacter roseus*	*Genu*	*Genus*	26482314 [88]
PJM	*Pseudoxanthomonas japonensis* MBL	*Gs*M	*Genus species* MBL	35943258 [119]
PLN	*Pedobacter lusitanus* NL19	*Gs*s	*Genus species* strain	30029312 [120]
POM	*Pseudomonas otitidis* MBL	*Gs*M	*Genus species* MBL	21060106 [121]
PNGM	Papua New Guinea MBL	LocM	Location MBL	29842976 [122]
RM3	Isolated from clone RM3	*Other*		27431213 [123]
SAM	*Simiduia agarivorans* MBL	*Gs*M	*Genus species* MBL	
SER	*Salmonella enterica* resistance	*Gs*r	*Genus species* resistance	32542533 [112]
SIE	*Sphingobium indicum* B3-E (E116)	*Gs*E	*Genus species* B3-E	34310207 [124]
SIQ	*Sphingobium indicum* B3-Q (Q116)	*Gs*Q	*Genus species* B3-Q	34310207 [124]
SMB	*Serratia marcescens* class B	*Gs*B	*Genus species* BBL	21876060 [125]
SPG	*Sphingomonas*	*Gen*	*Genus*	26482314 [88]
SPR	*Serratia proteamaculans*	*Gsp*	*Genus species*	23982345 [126]
SSE	*Sphingopyxis* sp. Enzyme?	*Gs*E	*Genus species* Enzyme	32542533 [112]
THIN-B	*Janthinobacterium lividum* class B	*Gen*B	*Genus* BBL	11181369 [127]

**Table 4 antibiotics-12-01746-t004:** Zn(II) ligands with consensus numbering in the three MBL subclasses.

MBL Subclass		Zn1 Ligands			Zn2 Ligands	
B1	H116	H118	H196	D120	C221	H263
B2	N116	H118	H196	D120	C221	H263
B3	H/Q/E116	H/R118	H196	D120	H/Q121	H/K263

**Table 5 antibiotics-12-01746-t005:** NDM renumbering algorithm [30]. The original numbering that corresponds to the amino acid position in a FASTA file is modified by adding the value in the Modification column to obtain the BBL numbering. This standard numbering follows the BBL standard numbering scheme and will accomplish renumbering of the Zn(II) ligands, among others, to the consensus numbers shown in Figure 1B and Table 4.

Original Number (FASTA Position)	Modification	BBL Number
1 to 110	−6	−5 to 104
111 to 134	−4	107 to 130
135 to 153	−3	132 to 150
154	−4 + a	150 a
155 to 201	+7	162 to 208
202 to 225	+13	215 to 238
226 to 239	+15	241 to 254
240	+14 + a	254 a
241	+13 + b	254 b
242 to 253	+13	255 to 266
254 to 270	+41	295 to 311

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
