# Peer review of "Strategies to Name Metallo-β-Lactamases and Number Their Amino Acid Residues"

_antibiotics, 2023, doi:10.3390/antibiotics12121746_

Round 1

Reviewer 1 Report

Comments and Suggestions for Authors

Metallo-β-lactamases (MBLs) are known to inactivate a broad range of β-lactams antibiotics; however, the naming of these enzymes is still not quite consistent in the field. Therefore, this review manuscript by Oelschlaeger et al. has revisited this standard numbering scheme and suggested some strategies on how its implementation could be made more accessible to researchers. This is a very timely and important contribution to the community of antibacterial research.

In general, the manuscript was properly organized and well-written and the references were cited properly as well. Therefore, the reviewer suggests the acceptance of the manuscript in Antibiotics as is. 

Reviewer 2 Report

Comments and Suggestions for Authors

It is meaningful to simplify the naming and numbering strategy for MBL. The authors summarized relevant content, which is helpful for other researchers. But how to unify the process and why it should be like this is quite important. The authors should emphasize the similarity and difference of different naming and numbering strategy and give a constructive solution instead of listing all the references.  

1. The authors compared MBL and SBL in the second paragraph of background, and it was suggested that a sketch map was listed in the paper in order to highlight the structural or property difference between the two.

2. There are two 2.2 as a title.

3. Please pay attention to the standard format of figure 2.

4. Table 5, detailed illustrations should be noted for researchers from different fields to understand the content.

Comments on the Quality of English Language

The language should be checked carefully for better understanding of the paper. For example, "It should be noted that other hydrolases exist that act on beta-lactam antibiotics outside the beta-lactam ring." It is difficult to follow the sentence.

Reviewer 3 Report

Comments and Suggestions for Authors

In the MS antibiotics-2721963, the authors wanted to retain the reader's interest by displaying the strategies to name metallo-beta-lactamases and number their amino acid residues. They have an interesting purpose, and their MS is descriptive; of 143 references, 44 were recently published after 2018. 

In the current version, the MS has the following sections and subsections:

1. Background

2. Naming of beta-lactamases

2.1. Based on the enzymatic reaction catalyzed

2.2. Naming of metallo-beta-lactamase families

2.2. Naming of metallo-beta-lactamase family members

3. Numbering amino acid residues in metallo-beta-lactamases

3.1. The class B (metallo-)beta-lactamase standard numbering scheme

3.2. Strategies for MBL renumbering

3.2.1. Manual renumbering

3.2.2. Automated renumbering based on conserved motive recognition

3.2.3. Automated renumbering based on profile Hidden Markov Model

The final ideas are placed in section 3, in a separate paragraph.

The following comments and suggestions are available below:

1. The abstract begins with an attractive background related to the NS content, and the authors briefly show the aim of their study.

2. The introduction is presented as a background by the authors.

This section begins with a short presentation of beta-lactam antibiotics, slowly leading to beta-lactamases as the most common antibiotic resistance mechanism (lines 24-38).

Then, the authors directly show metallo-beta-lactamases (MBLs) with their structure, classification, substrates, catalytic mechanisms, and potential inhibitors in an agglomerate presentation (lines 39-67).

In the following paragraph, the authors address the problem of naming MBLs and begin presenting their review's aim (lines 68-73).

More details are displayed in lines 74-116.

2.1. The reviewer considers that this introduction could be made more attractive to the readers. The authors are encouraged to expand the presentation of beta-lactamases, including classification, general structure, nominating the corresponding bacteria species, etc.

2.2. They can use suggestive schemes and figures, making their review more dynamic.

2.3. Moreover, they can try to display the most suitable arguments to convince the reader why knowing the strategies to name metallo-b-lactamases and numbering their amino acid residues is important.

2.4. The reviewer believes that the text from lines 74-116 could be a part of a new section, Discussion.

3. In Section 2, Naming of beta-lactamases, subsection 2.1. , the authors stated in line 148: "This review will focus exclusively on beta-lactamases (EC 3.5.2.6)". 

3.1. As the title indicates, this review focuses on metallo-beta-lactamases (class B). 

3.2. The authors are encouraged to check and restructure their MS content again. They can maintain section 2, with the same title, but perform a clear and schematic classification of beta-lactamases. In this mode, they can lead to class B metallo-beta-lactamases containing zincum.

3.3. Therefore, subsection 2.2. (Naming of metallo-beta-lactamase families) should become Section 3. The naming of metallo-beta-lactamases, with 2 subsections: 3.1. families, and 3.2. members.

4. Section 3 is entitled Numbering amino acid residues in metallo-beta-lactamases.

4.1. The authors are encouraged to make a suitable correlation between the previous sections and this one. Then, they could display all the information.

5. The previously mentioned Discussion section must correlate all data presented, make comparisons, discuss advantages/limitations, and indicate future directions and proposals.

Round 2

Reviewer 2 Report

Comments and Suggestions for Authors

It can be accepted.

Comments on the Quality of English Language

Check the language carefully, and the very very long sentence should be avoided.

Reviewer 3 Report

Comments and Suggestions for Authors

The reviewer considers that the authors suitably revised their MS according to all comments from the Round 1 Report.